

# Contribution of a constellation of two Wide-Swath Altimetry Missions to Global Ocean Analysis and Forecasting

Mounir Benkiran[1], Pierre-Yves Le Traon[1,2] and Gérald Dibarboure[3]

[1]Mercator-Ocean International, 31400 Toulouse, France

[2]Ifremer, 29280 Plouzané, France

[3]Centre National d'Études Spatiales, 31400 Toulouse, France

*Correspondence*: Mounir Benkiran (mbenkiran@mercator-ocean.fr)

**Keywords:** Data Assimilation; Ocean Forecasting; Surface Water Ocean Topography; Satellite Altimetry; Observing System Simulation Experiment

## Abstract

Swath altimetry is likely to revolutionize our ability to monitor and forecast ocean dynamics. To meet the requirements of the EU Copernicus Marine Service, a constellation of two wide-swath altimeters is envisioned for the long-term (post-2030) evolution of the Copernicus Sentinel 3 topography mission. A series of Observing System Simulation Experiments (OSSEs) is carried out to quantify the expected performances. The OSSEs use a state-of-the-art high resolution (1/12°) global ocean data assimilation system similar to the one used operationally by the Copernicus Marine Service. Flying a constellation of two wide-swath altimeters will provide a major improvement of our capabilities to monitor and forecast the oceans. Compared to the present situation with 3 nadir altimeters flying simultaneously, the Sea Surface Height (SSH) analysis and 7-day forecast error will be globally reduced by about 50%. With two wide-swath altimeters, the quality of SSH 7-day forecasts is equivalent to the quality of SSH analysis errors from three nadir altimeters. Our understanding of ocean currents is also greatly improved (30% improvements at the surface and 50% at 300 m depth). The resolution capabilities will be drastically improved and will be closer to 100 km wavelength compared to about 250 km today. Flying a constellation of two wide-swath altimeters thus looks to be a very promising solution for the long-term evolution of the Sentinel 3 constellation and the Copernicus Marine Service.





## 1. Introduction

The Copernicus Marine Service is one of the six pillar services of the European Union Copernicus programme (Le Traon et al., 2019). It provides regular and systematic reference information on the physical and biogeochemical ocean and sea-ice state for the global ocean and the European regional seas. After seven years of operation, the Copernicus Marine Service is recognized internationally as one of the most advanced service capabilities in ocean monitoring and forecasting, and has convinced more than thirty thousand expert services and users worldwide (Le Traon et al., 2021).

The Copernicus Marine Service is highly dependent on the timely availability of comprehensive satellite and in-situ observations (Le Traon et al., 2019). Satellite altimetry plays a prominent role thanks to global, real time, all-weather sea level measurements, which provide a strong constraint for inferring 4D ocean circulation through data assimilation (see a review in Le Traon et al., 2017). Copernicus Marine Service modelling and data assimilation systems depend substantially on the status of the altimeter constellation (e.g., Hamon et al., 2019). Four altimeters at least are required. The main limitation of classical altimetry is the 1D nature of altimeter measurements, which provide sea level measurements only at the sub-satellite point (i.e., nadir point), thus creating large unobserved gaps in the cross-track direction (Chelton et al., 2001). These authors also recall that the distance between altimetry satellite tracks and the revisit time decrease with the inverse of the number of satellites, so there is a strong diminishing return associated with classical altimetry. As a result, only wavelengths longer than 200 km are well represented.

Wide-swath altimetry that will be demonstrated with the Surface Water Ocean Topography (SWOT) mission (Morrow et al., 2019) addresses these limitations. Through a series of Observing System Simulation Experiments (OSSEs), Benkiran et al. (2021) and Tchonang et al. (2021) demonstrated that SWOT data could be readily assimilated in a global high resolution (1/12°) analysis and forecasting system with a positive impact everywhere and very good performances. The main limitation of SWOT is, however, related to its long-time repeat period. In the longer run, flying a constellation of two wide-swath altimeters would thus be highly beneficial to further improve performance, in particular, for smaller space and shorter time scales.

The impact of a constellation of two wide-swath altimetry missions has been analysed as part of two studies carried out by Mercator Ocean International (MOi) for the European Space Agency (ESA). This was done in close collaboration with the French Space Agency (CNES), which led a 2.5 year (2018-2020) phase A study called WiSA (Wide-Swath Altimetry). WiSA aimed to define an innovative concept of altimetry system to provide new measurements for both oceanography and hydrology on an operational basis (CNES, 2020). The targeted programmatic framework is the Sentinel-3 Topography mission (post 2030), the follow-up to Sentinel-3, which will address the expected evolution of the Copernicus Space Component. The first ESA study carried out by Bonaduce et al. (2018) in the North East Atlantic regional model showed that a constellation of three nadir and two wide-swath altimeters could reduce ocean analysis errors by up to 50% compared to three nadir altimeters




operating alone. The second ESA study (this paper) extends this work to the global ocean using the wide-swath
altimeter system characteristics analysed as part of the WiSA phase A. The study also uses the latest version of
the Copernicus Marine Service global 1/12° modelling and data assimilation system and the OSSE design used
for SWOT OSSEs presented in Benkiran et al. (2021) and Tchonang et al. (2021).
Results for this global study are presented and discussed in this paper, which is organized as follows. The WiSA
concept is presented in section 2. Section 3 details the OSSE methodology. Results are discussed in Section 4 and
Section 5 provides the main conclusions and recommendations of the study.
**2. The WiSA concept**
The WiSA concept was developed in a Phase A study carried out by CNES and the industry as a tentative follow-
up to the SWOT mission. The goal of the WiSA concept was to leverage the main improvement of SWOT's swath
altimeter (i.e., 2D images of sea surface topography and near-nadir radar backscatter, lower noise floor than nadir
altimeters for the same ground pixel surface) with significant changes to better address the needs of operational
oceanography and hydrology, while making the satellite simpler, smaller and more affordable than the SWOT
precursor mission.
Arguably the main weakness of SWOT is temporal sampling: with a single satellite and a 120-km swath, it is
simply impossible to resolve the time scales of the small-scale features that will be observed by SWOT. At least
two (resp. three) wide-swath altimeters are needed to ensure that 68% (resp. 80%) of 50 km features in the global
ocean can be observed with a mean revisit time of 5 days or less. Moreover, Lamy and Albouys (2014) explain
how the SWOT orbit was a trade-off between multiple constraints for this research mission: technical constraints
from the instrument, optimization for the aliasing of tides, sampling optimized for a single satellite. In contrast,
the so-called WISA #A orbit was selected by CNES using the methodology of Dibarboure et al. (2017) to
maximize the sampling for 1 to 3 swath altimeter satellites (or swath/nadir hybrid constellations). This sun-
synchronous orbit has an altitude of approximately 750 km (14+7/17 revolutions per day) and the altimeter swath
covers latitudes up to 82°. With an exact repeat of 17 days (i.e., good enough for tidal aliasing despite being sun-
synchronous), the orbit also has so-called sub-cycles of 2 and 5 days, both of which maximize the distribution of
observations in space and time for wind/wave applications and small to medium mesoscale applications.
Moreover, the optimal space/time sampling for two satellites is achieved when the first and second swath altimeters
are on the same orbit plane, separated by a 180° angle on the orbit circle. This property, discussed by CNES (2020),
is important for technical and practical considerations (e.g., ground station visibility, compatibility with other
sensors with a wider swath).

The second difference between SWOT and WiSA is their respective noise levels. SWOT tries to achieve an
unprecedented spectral noise floor of 2 cm²/cycle/km (i.e., of the order of 1.36 cm RMS for 2 km x 2 km pixels)
in order to resolve wavelengths as small as 15 km (i.e., a feature diameter of 8.5 km). Because the goal of WiSA
is to resolve wavelengths of only 50 km, CNES and Thales Alenias Space selected a simpler technical design (i.e.,
cheaper and more robust) for the interferometer baseline. The resulting noise is of the order of 2.7 cm RMS for 2



km x 2 km products. Note that because the high-frequency noise is random, it can be averaged out in pixels of
varying size. In other words, it is possible to select different resolution-versus-precision trade-offs from the same
spectral noise floor (e.g. sub-centimetric precision for a 5 km product or 5 cm precision for a 1 km product). More
importantly, while larger than SWOT, the noise floor of the WiSA concept is more than sufficient to ensure that
all scales up to 50 km are well observed, even in relatively high wave conditions (wave height modulates the noise
floor of all altimeters according to a linear relationship). Using the methodology of Vergara et al. (2019) and
spectral slopes and wave climatologies obtained from Jason altimeters, CNES (2020) reports that WiSA has a
mean observability (i.e., wavelength where the signal to noise ratio is 1) of 37 km or better, over more than 80%
of the global ocean.
**3.  OSSE approach**
**3.1 Ocean Model**
The MOi global ocean forecasting system (see Lellouche et al., 2018), which delivers forecast products for the
Copernicus Marine Service, is used in this study. As described in the literature (Errico et al., 2013), OSSEs use
two different models or model configurations. In our study, we use the same NEMO 1/12° resolution model
(Nucleus for European Modelling of the Ocean, Madec, 2016) but with different configurations and forcings. The
first uses a free NEMO3.6 simulation (Nature Run hereafter referred to as NR) to represent the real ocean and
simulate all the synthetic observations for the study. The second model is used to assimilate synthetic observations
from the NR in a so-called Free Run (FR). The FR uses the NEMO3.1 model with a different and less energetic
configuration. A detailed description and validation of the NR is presented in Benkiran et al. (2021).
**3.2 Simulation of observations and noise**
All simulated observations were extracted from the NR simulation and these observations were collected over a
period of 15 months (from October 1, 2014 to December 31, 2015), which includes the period covered by the
OSSEs. Sea Surface Height (SSH) data were simulated along 3 nadirs and 2 wide-swath altimeters (S1 and S2).
The 3 nadir altimeters correspond to Jason-3 (or Sentinel 6, which will use the same orbit) (J3) and the nadirs of
each of the two Wide-Swath Altimeters S1 and S2. The along-track nadir altimeter data were extracted from NR
at the 1 Hz frequency corresponding to a spatial resolution of 6-7 km from hourly mean fields of the NR. A random
noise of 3 cm was added to along-track data to take into account altimeter measurement noise (i.e., close to the 1
Hz error budget of the nadir altimeter of the SWOT satellite). For the two swath altimeters, the WiSA #A orbit
(S1) selected by CNES (Dibarboure et al., 2017) was used together with a second (S2) on the same orbit plane,
separated by a 180° angle on the orbit circle. All SSH data were simulated from the NR using the Jet Propulsion
Laboratory's (JPL) SWOT Simulator (Gaultier et al., 2016). The simulator constructs a regular grid based on the
baseline orbit parameters of the satellite. The simulator models the most significant errors that are expected to
affect the data, i.e., the KaRIn (Ka-band Radar Interferometer) noise, roll errors, phase errors, baseline dilation
errors, wet troposphere and timing errors. In this study, we only used the estimated WiSA KaRIn noise derived
for a significant wave height (SWH) of 2 m. Figure 1 shows the standard deviation of the KaRIn random error





considering across-swath resolutions of 1 km (solid line) and 6 km (dashed line) as a function of the cross-track
distance in km.
Figure 2A shows the SSH from the NR at a given central date of our 7-day assimilation cycle (analysis window)
over the Kuroshio region. Data coverage along the tracks of the three nadir altimeters over the 7-day analysis
window is shown on Figure 2B while Figure 2C shows the coverage of the combination of two Wide-Swath
Altimeters. It can be observed that with two Wide-Swath Altimeters the ocean is almost covered by the
measurements over a 7-day time period.
To make OSSEs close to the MOi global analysis and forecasting system, other assimilated data were simulated:
satellite sea surface temperature, in-situ temperature and salinity data (Argo) and the ice concentration data (see
details in Benkiran et al., 2021).
**3.3 Data Assimilation**
An updated version of the data assimilation scheme developed at MOi, called SAM2 (*Système d'Assimilation*
*Mercator* V2) and described by Lellouche et al. (2018), was used. SAM is a reduced-order local Kalman filter for
which the analysis subspace is constructed using a band-passed times series of model states from free simulation.
Several improvements and adaptations of this system were made for this study. In particular, a four-dimensional
(4D) version of the assimilation scheme is used, in which the analysis uses a 4D subspace and produces daily
model correction of SSH, temperature, salinity and velocities fields. All these updates and their impacts on the
system performance are described in Benkiran et al. (2021).
**3.4 Experimental set-up**
Starting from the satellite altimetry simulated data obtained from the NR run, three global OSSEs were carried out
using a different NEMO configuration but the same spatial resolution of 1/12° (~ 7 km). OSSE0 is the Free Run
(FR) of the ocean model used to assess the performance of the other experiments. OSSE1 corresponds to nadir
(3N) altimetry data assimilation. OSSE2 (not presented here) is similar to OSSE1 except that it assimilated Sea
Surface Height (SSH) from two Wide-Swath Altimeter (2S) datasets instead of nadir altimeter data. Finally,
OSSE3 (3N+2S) assimilated all observation types (combining two swaths and three nadirs). OSSE1, 2 and 3 also
assimilate Sea Surface Temperature (SST), Ice Concentration (IC), and Temperature and Salinity (T/S) profiles
data.

Our simulations start from a free model state on October 1, 2014. A three-month simulation (until the end of
December 2014) was carried out with assimilation of SSH along the Nadirs (3N) together with SST, IC and T/S
data. This allows us to avoid the spin-up period in our experiments. All the experiments shown here start from the
same state on January 1, 2015.
**4.  Results**
Results of the impact of assimilation of the SSH from the three nadir altimeters and from the two wide-swath
altimeters combined with the three nadir altimeters are detailed in this section. These results are obtained by

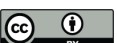



comparing each experiment with our real ocean (NR) data over a period of 10 months (March 1st to December 30th
2015). Results are presented below: impact on SSH analyses and forecasts, impact on the different time and space
scales, spectral and coherence analyses in a series of selected rectangular areas (boxes) and impact on velocity,
temperature and salinity.

**4.1 Impact on sea-level analyses and forecasts**

The SSH variance in the NR computed over one year (2015) is shown on Figure 3. The SSH variance shows a
high variability in the more energetic regions such as the Gulf Stream (GS), Kuroshio (KS), Antarctic Circumpolar
Current (ACC), Brazil-Malvinas confluence (BM) and Agulhas (AG). The SSH variance in the NR compares very
favourably with real altimeter observations (as detailed in Benkiran et al., 2021).
The temporal evolution of SSH variance error over the global ocean for each experiment is compared in Figure 4.
This variance decreases over a few weeks (6 weeks) to reach a stable state for the analyses (continuous lines) and
the forecasts (dotted lines). The assimilation of the swath altimeter data reduces analysis errors from 15.6 cm$^2$
(black line) to 10.1 cm$^2$ (red line), a reduction of 54%. The gain is of about 46% for the forecasts (dotted lines).
These are major improvements. In particular, with two swath altimeters, 7-day SSH forecasts are as good as SSH
analyses derived from three nadir altimeters.
To analyse the results further, the relative variance $VAR^*$ (in terms of percentage), which represents the ratio of
the variance error (VarError) to the variance of the NR signal, was computed. This variance of the error (VarError)
is calculated by comparing each OSSE with the NR.
Global maps of SSH analysis error variance (VarError) for these experiments are presented in Figure 5. The Free
Run (FR) has a fairly large variance (Figure 5A) especially over western boundary currents: Antarctic Circumpolar
Current (ACC), Indian Ocean (IND), Brazil-Malvinas confluence (BM) and Agulhas current (AG). The
assimilation of SSH data from the 3 nadir altimeters (3N) in addition to the sea surface temperature data and
salinity and temperature profiles significantly reduced this error (Figure 5B) over the global ocean. The
assimilation of SSH from the swath altimeter in addition (Figure 5C) greatly reduced this error over the global
ocean. A fairly significant improvement can be observed in specific areas (boxes in Figure 5B) such as the Gulf
Stream, the Antarctic Circumpolar Current and the Kuroshio Current. Figure 6, which represents the difference in
error between the 3N assimilation and the 3N+2S assimilation, shows the contribution of the assimilation of the
combination of the Wide-Swath Altimeters and the Nadir altimeters compared to the Nadir altimeters. This
improvement (error reduction) is visible on almost 80% of the ocean points. The results of the impact of the Wide-
Swath Altimeter data on SSH in global ocean are summarized in Table 1. Adding the swath altimeters improves
the analyses and forecasts by about 50%. With the assimilation of wide-swath altimeters, the relative error ($VAR^*$
- Columns 3 and 4 in Table 1) is only 14 and 21% for analyses and forecasts respectively.

To better quantify the impact of swath data in the global system, errors are characterized for specific time and
space scales. Figure 7 compares the error variance of the different OSSEs for wavelengths smaller than 200 km.
The assimilation of nadir altimeters (Figure 7B) already considerably reduces these errors compared to the FR.



The addition of the swath altimeter assimilation (Figure 7C) brought a major improvement for the regions where
large errors remained (small scales) with the assimilation of nadir data. This improvement is prominent in the Gulf
Stream, Kuroshio and ACC.
Figure 8 similarly compares these errors for periods of less than 20 days. The aim is to analyse the impact of Wide-
Swath Altimeter data on fast signals. Much of the error is corrected by assimilating the swath data in addition to
the nadir data (Figure 8C) compared to the nadir data alone (Figure 8B). This improvement is visible throughout
the global ocean. This shows that there is better control of the high frequency signals by assimilating swath
altimeter data as opposed to nadir data.
Figure 9 summarizes the main results of these analyses. The three panels of Figure 9 represent the mean error
variance as a function of latitude for the total error, the error for wavelengths < 200 km and the error for periods
< 20 days. The swath altimeter data assimilation reduces the error at each latitude (red curves on the panels). This
improvement is more pronounced at mid and high latitudes than at low latitudes. The impact on the western
boundary currents and ACC currents is more evident at the mesoscale (< 200 km) than at frequencies below 20
days.
**4.2 Spectral analysis and coherence**
Wavenumber Power Spectral Density (PSD) and spatial and temporal coherence for each OSSE are discussed in
this section. Spectra and coherence on boxes covering 10° in latitude by 20° in longitude at different latitudes
(boxes in Figure 3) are computed. Spectra were also computed on the same box (North Atlantic Drift: 19°W,10°W;
46°N,55°N, Box D in Figure 5B) as that presented by Bonaduce et al., 2018, using a regional model (IBI: Iberian-
Biscay-Irish region) to make comparisons.
Figure 10 shows the power spectra of the SSH error in a variance preserving form (Thomson and Emery, 2014) in
the different boxes. The assimilation of nadir altimeter data (black curves) reduces the error at different scales
compared to the FR (orange curves) except in region A (low-latitude, Figure 10A) where the assimilation of nadir
data introduces noise in the 50-200 km wavelength band. This is mainly due to the weak signal in these regions
and the limited space/time sampling of the nadir altimeter constellation at these wavelengths. The assimilation has
difficulty extrapolating the small-scale structures between the tracks (see discussion in Section 3.4.1 of Lellouche
et al., 2018). The contribution of the swath altimeter data contributes to a clear reduction in the error spectra in all
these regions for wavelengths larger than 50 km.

The reduction of the error at the different wavelengths ($ER_{spec}$) is defined as the percentage of the error with respect
to FR (OSSE0). In all these areas (boxes) the assimilation of nadir altimeters (black curves) reduces the $ER_{spec}$
error by more than 65% between 200 and 600 km. A major contribution is observed with swath altimeter data, as
error reduction exceeds 90% between 200 and 600 km over these regions (red curves, $ER_{spec}$ > 90%).

Spectral coherence analysis (temporal and spatial) is also performed to highlight the impact of assimilating swath
data at different scales with respect to the NR. Coherence is defined as the correlation between two signals as a





function of wavelengths (Ubelmann et al., 2015; Ponte and Klein, 2013; Klein et al., 2004). This coherence
between the NR and different OSSEs is defined as follows:

$$C_{spec} = \frac{Cr_s(NR,\ OSSE_j)}{S(NR)\ S(OSSE_j)}$$

Where $Cr_s$ $and$ $S$ represent the cross-spectral density and spectral density, respectively of the signals $j$ referring $j$
-the experiment. The impact of the swath data is clear over all these regions (different latitudes) from 50 km of
wavelength (red curves on the Figure 11). The wavelengths and periods with a coherence of 0.5 (dotted line on
the figures), which are usually taken as an estimation of the effective resolution (e.g., Ubelmann et al., 2015;
Tchonang et al., 2021), show that wide-swath altimeter data (red lines) will provide much improved insight into
mesoscale ocean dynamics as compared with nadir altimeter data (black lines). In box D (Figure 11D), which
represents the North Atlantic Drift (the same box as that presented in Bonaduce et al., 2018), the wide-swath
altimeter gain in the effective resolution is in the region of 60% (105 km instead of 165 km for nadir altimeters).
At low latitudes (box A, Figure 11A), there is an improvement of around 80% thanks to wide-swath altimeter data.
This improvement is also observed at high latitudes (Figure 11B and Figure 11C) but it is less pronounced.
Figure 12 shows the time coherence for the four selected regions. The calculation of this coherence was based on
filtered SSH fields of scales greater than 500 km to avoid the impact of large-scale and high frequency signals on
the results. At different latitudes (regions shown), wide-swath altimeter data improved the temporal coherence
with the NR compared to the nadir data. The effective time resolution in regions A and B (Figure 12A and Figure
12B) is 20 days for wide-swath altimeters instead of 40 days for nadir altimeters (half the time). At mid and high
latitudes (Figure 12C: Gulf Stream and Figure 12D: North Atlantic Drift), there is a strong improvement, with a
time resolution of 25 days for wide-swath altimeters whereas with the nadir altimeter data the coherence never
exceeds 40%.
**4.3 Impact on temperature, salinity and zonal velocities**
Figure 13 shows the variance of the temperature and salinity error (NR-OSSEs) as a function of depth for the
global ocean. The temperature error profile shows a maximum at about 100 m depth, which represents the
thermocline. This error is significantly reduced by assimilating the nadir altimeter data (black profile) compared
to the free model (FR, orange profile). The assimilation of the Wide-Swath Altimeter data does not degrade this
score and we even have a slight improvement between 100 m and 750 m depth. For salinity (Figure 13B), the
improvement is less clear, but no degradation is observed at any depth.

Figure 14 similarly shows the average variance error both of the zonal (U) and meridional (V) velocity as a function
of depth for the global ocean for each of the experiments. The assimilation of nadir altimeter data (black profiles)
brings a significant reduction of this error with respect to the FR (orange profiles) between the surface and 1000
m depth on both U and V. There is a clear reduction over the whole depth with the wide-swath altimeter data
assimilation (red profiles) on both velocity components    Similarly, Figure 15 compares the evolution of the



velocity error variance as a function of time over the year 2015 at the surface and at 300 m depth. On the surface
(top panels), there is a constant improvement (red curves) on both components (U and V). On the other hand, at
300 m depth, there is a reduction that sets in after one month and remains constant over the year. Table 2
summarizes the statistics on the horizontal velocities from Figure 15. Overall, there is an error reduction of more
than 30% for the surface currents and more than 50% for the currents at 300 m with Wide-Swath Altimeter data.
Wide-swath altimeter (2D) data allow a much better constraint of the ocean dynamics compared to the assimilation
of nadir data.
**5.   Summary and conclusions**
The SWOT mission to be launched at the end of 2022 will demonstrate the potential of swath altimetry, which is
likely to revolutionize our ability to monitor and forecast ocean dynamics from mesoscale to submesoscale. SWOT
will considerably improve on the capabilities of the present constellation of nadir altimeters (Benkiran et al., 2021;
Tchonang et al., 2021) but its time sampling (21 days) will be a limitation. A constellation of two Wide-Swath
Altimeters will provide, however, much better space/time sampling and should allow us to observe 68% of the
ocean every 50 km and 5 days (CNES, 2020). Such a configuration is envisioned by ESA for the long-term
evolution (post-2030) of the Copernicus Sentinel 3 topography mission to meet the requirements expressed by the
Copernicus Marine Service and its applications (CMEMS, 2017). To quantify the expected performances, a series
of OSSEs have been carried out in this study using a state-of-the-art high resolution (1/12°) global ocean data
assimilation system.
Results confirm the high potential of such a configuration. Flying a constellation of two wide-swath altimeters
will provide a major improvement of our capabilities to monitor and forecast the oceans. Compared to the present
situation with 3 nadir altimeters flying simultaneously (Sentinel 6 and the two Sentinel 3), the SSH analysis and
7-day forecast error will be globally reduced by almost 50%. Improvements will be much larger in mid and high
latitude regions and less in tropical/equatorial regions. Surface and deep velocity fields will also be greatly
improved. Surface current forecast errors should be equivalent to today's surface current analysis errors or
alternatively will be improved (variance error reduction) by 30% at the surface and 50% for 300 m depths.

The resolution capabilities will be drastically improved and will be closer to 100 km wavelength as opposed to
today where they are above 250 km (on average). On average, on the four boxes presented (representative of
different latitudes), there is a 60% improvement of the resolved structures with the two wide-swath altimeters. In
terms of time scales resolved, improvements will be larger than expected for time periods around 20 days (50% of
coherence, improvements of 60%).

Flying a constellation of two wide-swath altimeters thus looks to be a very promising solution for the long-term
evolution of the Sentinel 3 constellation and the Copernicus Marine Service.



1 **Acknowledgments:** The study was funded by ESA and was also carried out as part of a partnership agreement

2 between Mercator Ocean International and CNES. All participants in the WiSA study are thanked for their high

3 quality work.



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

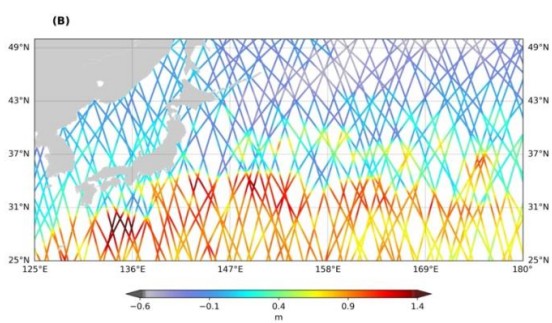

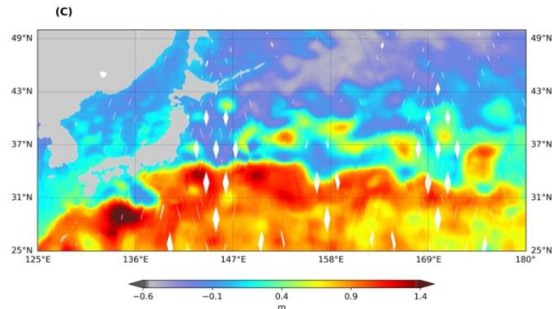

**Figure 2**: (A) SSH from Truth Run (NR) on 4th January 2015, (B) simulated along-track data from Jason3, nadirs of S1 and
S2 for seven-day assimilation cycle, and (C) simulated S1+S2 data (01/01/2015-08/01/2015).





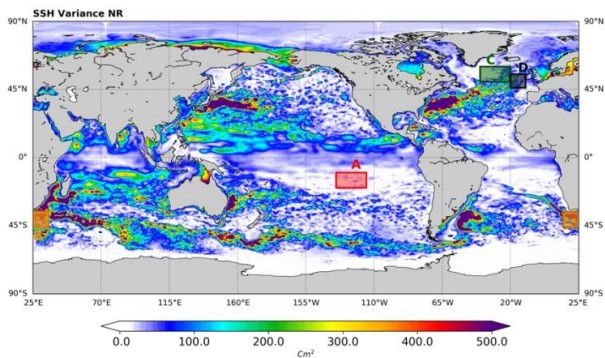

**Figure 3**: SSH variance (in cm²) in the NR over the period from February to December 2015. The boxes denote the rectangular
sub-regions for which wavenumber spectra and coherence analyses were performed.

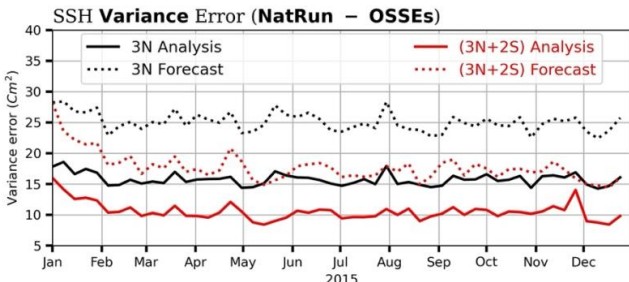

**Figure 4**: The temporal evolution of the SSH error variance in global ocean analysis and forecast over 2015. Results obtained
by comparing the SSH ocean analysis (solid lines) and forecast (dash lines) with the SSH from the NR. Experiments with
assimilation of 3 nadirs (3N): black lines and with assimilation of 3 nadirs and 2 Swath altimeter (3N+2S): red lines.


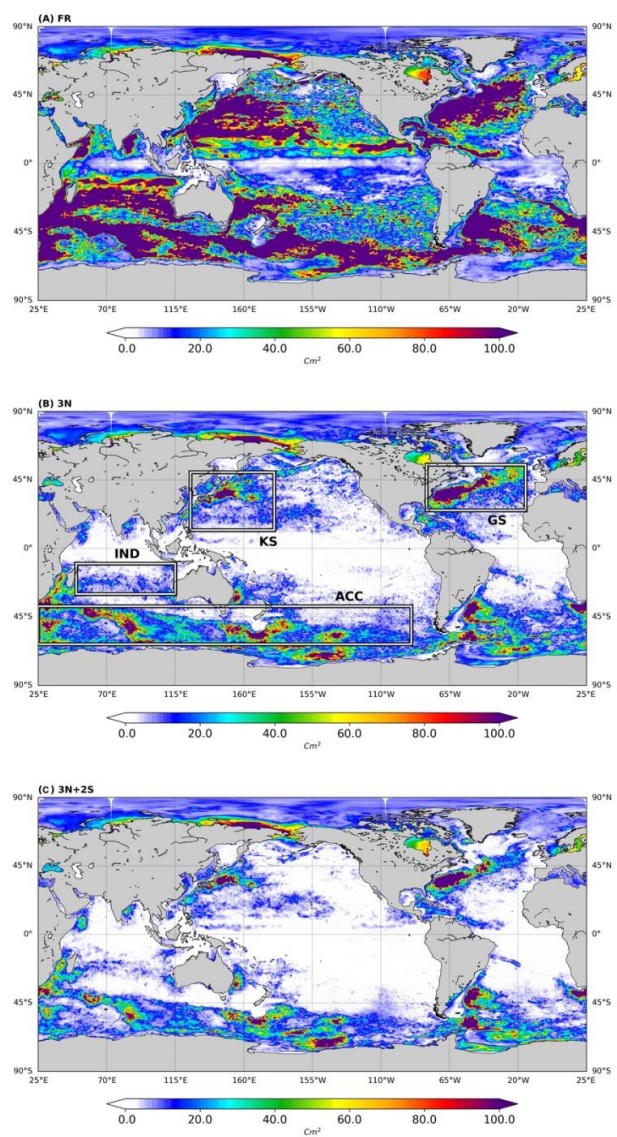

1 **Figure 5**: Global Maps of SSH analysis error (NR – Model Analysis) variance (in cm², 2015). (A) Free Run (FR); (B) With 3

2 nadirs (3N); (C): Assimilation of 3 nadir and 2 Wide-Swath (3N+2S). In panel (B), GS: Gulf Stream, ACC: the Antarctic

3 Circumpolar Current, KS: Kuroshio Current and IND: South Indian Ocean.



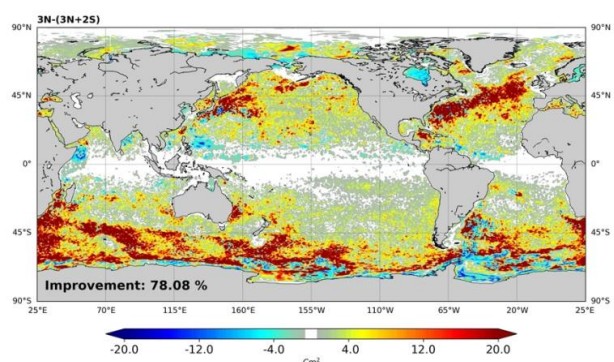

**Figure 6**: Difference between analysis error variance of assimilation with 3 nadirs (3N) and assimilation of 3 nadirs and 2
Wide-Swath Altimeter (3N+2S).

| | VarError (cm$^2$) | | $VAR^*$ (%) | |
|---|---|---|---|---|
| | Analysis | Forecast | Analysis | Forecast |
| OSSE1 (3N) | 15.6 | 24.8 | 21.2 | 30.4 |
| OSSE3 (3N+2S) | 10.1 | 17.0 | 14.1 | 21.3 |
| Gain | 54% | 46% | 50% | 42% |

**Table 1**: SSH ocean analysis and forecast error statistics during the year 2015. Columns 1 and 2 represent the analysis and
forecast variance of error computed from the difference between the OSSE and the NR (VarError, cm$^2$). Columns 3 and 4 show
the ratio of the variance of the error relative to the Nature Run variance (Var*, %).



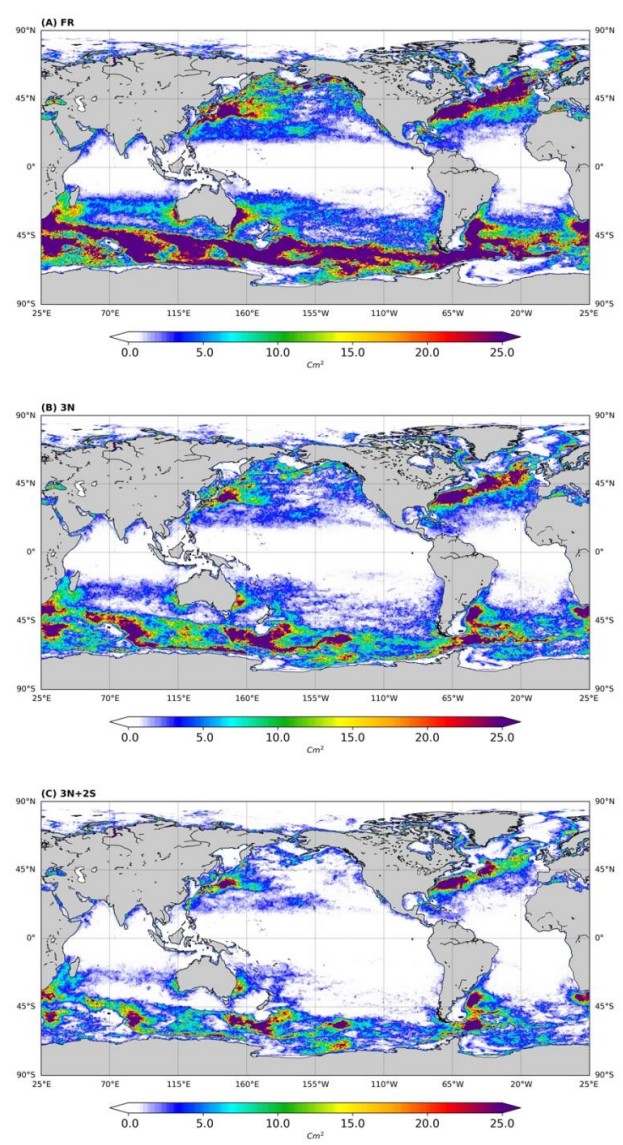

1 **Figure 7**: Global maps of SSH analysis error (NR – Model Analysis; Wavelengths < 200km) variance (in cm², 2015). (A) Free
2 Run (FR); (B) With 3 nadirs (3N); (C) with 3 Nadir and two Wide-Swath (3N+2S).

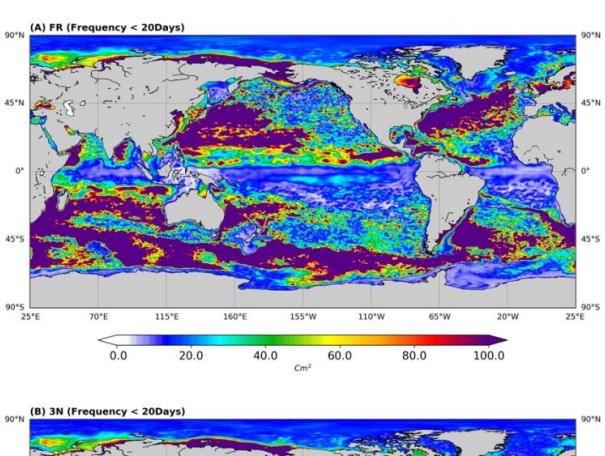

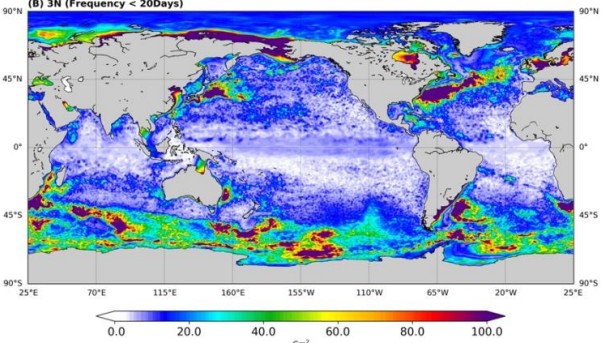

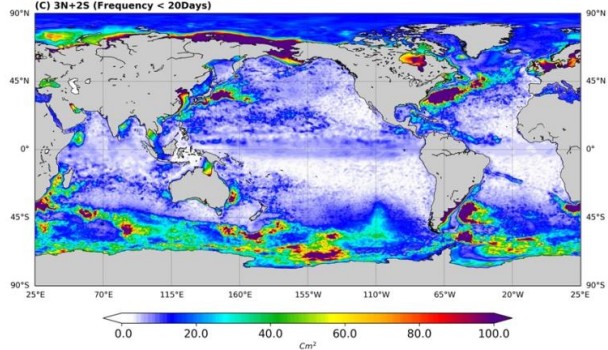

2  **Figure 8**: Global maps of SSH analysis error (NR – Model Analysis; Time scales < 20Days) variance (in cm², 2015). (A) Free

3  Run (FR); (B) With 3 nadirs (3N); (C) with 3 nadir and two Wide-Swath (3N+2S).




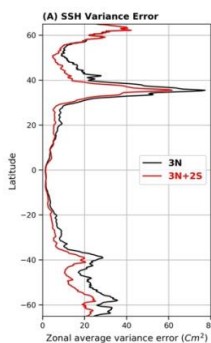
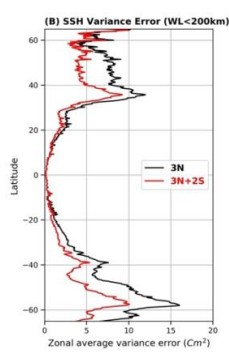
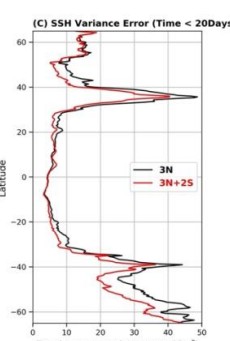

**Figure 9**: The zonal averaged error variance of SSH: (A) for full scales, (B) for scales less than 200 km and (C) for time scales
less than 20 days; assimilation of 3N (black lines) and assimilation of 3N+2S (red lines). Units are $cm^2$




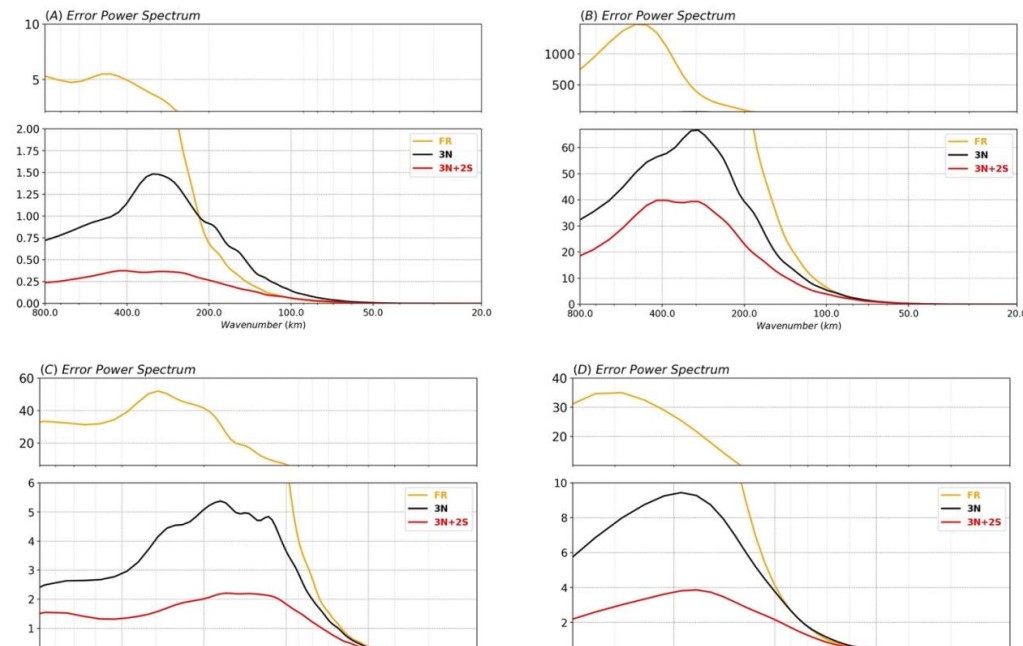

**Figure 10**: Power spectra SSH Error with respect to the NR; the spectra are shown in a variance preserving form (cm$^2$), (A)
low-latitude region (red box in **Figure 3**), (B) Agulhas current (orange box in **Figure 3**), (C) North Atlantic (high-latitude,
green box in **Figure 3**) and (D) North Atlantic Drift current (black box in **Figure 3**).




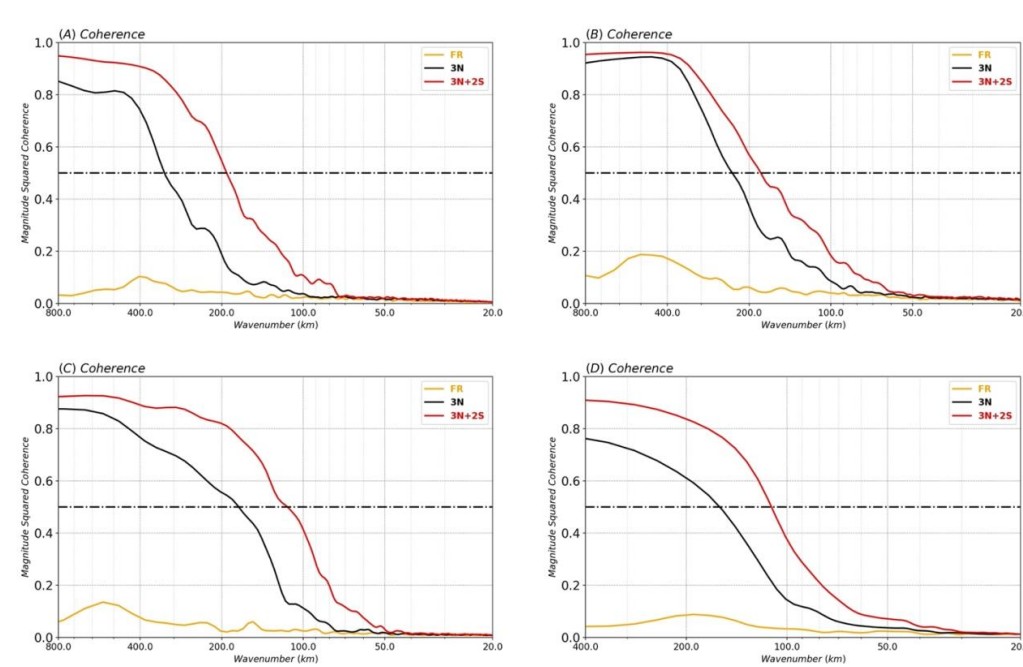

2 **Figure 11**: Wavenumber spectral coherence with respect to the NR, (A) low-latitude region (red box in **Figure 3**), (B) Agulhas

3 current (orange box in **Figure 3**), (C) North Atlantic (high-latitude, green box in **Figure 3**) and (D) North Atlantic Drift current

4 (black box in **Figure 3**).




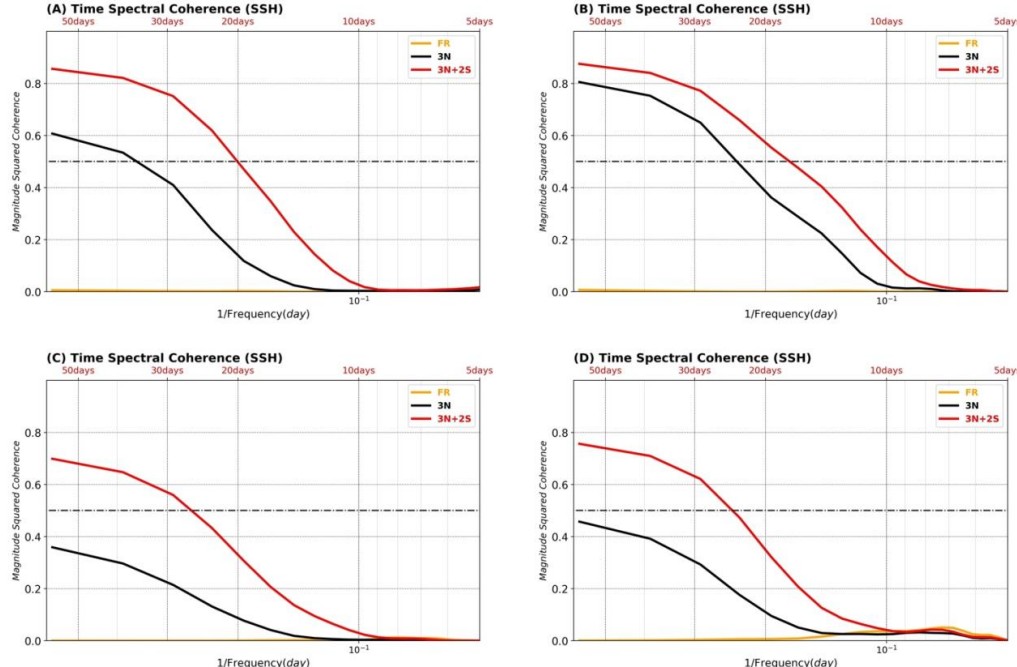

**Figure 12**: Time spectral coherence with respect to the NR (Wavelengths < 500km), (A) low-latitude region (red box), (B)
Agulhas current (orange box), (C) North Atlantic (high-latitude, green box) and (D) North Atlantic Drift current (black box).




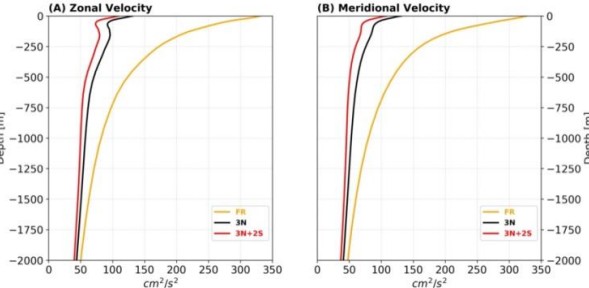

**Figure 13**: Global averaged error variance: (A) Temperature (in Deg$^2$) and (B) Salinity (Psu$^2$) over the period of March to
December 2015. The results were obtained by comparing the zonal and meridional velocities of OSSEs with the NR; FR orange
lines, 3N black lines and 3N+2S red lines.
**Figure 14:** Global averaged error variance (in cm$^2$/s$^2$): (A) zonal velocity and (B) meridional velocity over the period of March
to December 2015. The results were obtained by comparing the zonal and meridional velocities of OSSEs with the NR; FR
(orange lines), OSSE1 (3N; black lines) and OSSE3 (3N+2S; red lines).





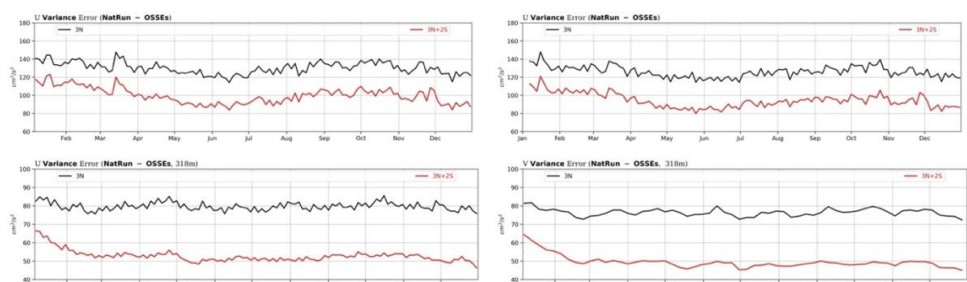

**Figure 15**: Temporal evolution of zonal and meridional velocity error variance (cm²/s²) for 7-day ocean analysis over the
period from 1 January to December 20, 2015. The results were obtained by comparing both components of the velocity (U, V)
to that of the NR. Black curves with assimilation of nadir altimeters (3N) and red curves with assimilation of nadirs and wide-
swaths (3N+2S). Upper panels for surface velocities and lower panels for velocities at 300 m depth. The statistics are
summarised in Table 2.

|  | VarError (cm²/s²) | | | |
|---|---|---|---|---|
|  | $U_{Surface}$ | $V_{Surface}$ | $U_{300m}$ | $V_{300m}$ |
| OSSE1 (3N) | 130.2 | 125.7 | 79.7 | 76.5 |
| OSSE3(3N+2S) | 99.4 | 94.1 | 52.7 | 49.5 |
| Gain | 31% | 33% | 50% | 53% |

**Table 2**: Statistics for velocities at surface and 300 m depth