# Peer review of "Contribution of a constellation of two Wide-Swath Altimetry"

_Ocean Science, 2021_

## Referee Comment (RC1)

Review of "Contribution of a constellation of two Wide-Swath Altimetry Missions to Global Ocean Analysis and Forecasting" by Mounir Benkiran, Pierre-Yves Le Traon, and Gérald Dibarboure

The manuscript is generally well written and understandable. The authors present results of several ocean prediction OSSEs to simulate the impact of two swath altimeter satellites in combination with three nadir altimeter satellites. The results are well presented. The results are significant and will be of interest to the science community for determining the importance of future investment. The quality of the work is high. The results are based on realistic ocean forecast systems being applied operationally. These results build on many prior studies on the impact of satellite and in situ observations. Most the comments I have are for improving the clarity of text and presentation. There are a few points that may require further consideration to resolve, though it is certainly possible I did not understand the intent of the text clearly, and in this case it may be relatively easy to resolve. The significant points are highlighted in yellow below.

7: "and has convinced more than thirty thousand expert services and users worldwide" – Do you mean that there are 30,000 users worldwide?

18: "so there is a strong diminishing return associated with classical altimetry" – we have seen a linear improvement in spatial correlation in properties associated with fronts. Surface divergence and frontogenesis forcing in particular increase linearly from 1 to 4 nadir altimeters. So the statement on diminishing returns depends on which properties are examined. For ssh, the statement is certainly correct. Clarification would be helpful.

P3, 17 – "it is imply impossible to resolve the time scales of the small-scale features" That is a bit of a strong statement that could use some context. It applies to submesoscale instabilities in the open ocean. Arguably, tidal variability in coastal areas will be resolved in space, and though aliased in time SWOT observations should provide new information on small scale tides. Help the reader understand the particular ocean features to which the statement applies.

P3, 20 – "can be observed with a mean revisit time of 5 days or less" – What is the source of the 5 day limit? Energetic areas such as the Gulf Stream or Kuroshio exhibit large changes even over 3 day time scales.

P3, 26 – "good enough for tidal aliasing despite being sun-synchronous" – That depends on application. It is not necessarily good enough for diurnal tides.

P3, 35 – "spectral noise floor of 2 cm²/cycle/km" – The SWOT requirement should be stated clearly. Leaving the statement as this leaves the reader to assume this applies in the along track and cross track directions and that the noise is uncorrelated. The correlated error levels are very important. These have specific correlation shapes in the across track direction with spectral power that is not white noise in the along track direction.

P4, 14 – "In our study, we use the same NEMO 1/12° resolution model but with different configurations and forcings" – It would be helpful for readers to have a few additional details on the model system at this point. What is the vertical resolution? This would help inform the reader as to the realism of the vertical structure of features in the nature run. What surface forcing is used at what temporal frequency? Is tidal forcing used in the system? What is the turbulence closure representation? Since the nature run is a free run, what is the realism of the circulation and energetics?

P4, 25 – "or Sentinel 6, which will use the same orbit" – verb tense. Sentinel-6A already is in the same orbit.

P4, 31 – "All SSH data were simulated from the NR using the Jet Propulsion Laboratory's (JPL) SWOT Simulator" – It was stated earlier that the WiSA design is simple and more cost efficient than SWOT though could resolve features only at larger scales. What is changed in the SWOT simulator to account for these differences? Are there any differences in correlated errors in the system? Will the predominant correlated errors still be baseline, roll, phase, and timing?

P4, 35 – "In this study, we only used the estimated WiSA KaRIn noise derived for a significant wave height (SWH) of 2 m" – How might correlated noise affect the results? These are significant in the roll, phase, timing, and baseline errors of SWOT. Correlated noise usually requires thinning or decimation of the data to the point where observation noise is uncorrelated, or the observation noise cross covariance matrix should account for this explicitly.

P5 – 17 – "All these updates and their impacts on the system performance are described in Benkiran et al. (2021)." – Details would help the reader here. There is some contradiction with the information on lines 3-4 of page 5 stating a 7 day assimilation cycle versus the statement of 1 day. Though details are in Benkiran, it would be helpful to have some basic information here. What is the frequency at which the analysis is performance (daily or 7 days)? What is the data interval for each analysis (the prior 24 hours, prior 7 days, a 7 day period centered on the analysis time, …)? Is the data thinned or decimated in the processing for the analysis? For a 1/12 degree model that represents features on the order of 45 – 90 km wavelength, the 6 km sampling of the WiSA data produces greatly redundant information. How is this handled? If the assimilation system uses a covariance produced from model free run, and the observations are from a model free run, then the simulated data will be very consistent with the assimilation system assumptions within the covariances. How will this affect the results in the real world if the assumed covariances are not accurate? Will a WiSA still produces the same level of impact concluded in the paper?

P5 – 23 – "OSSE2 (not presented here) is similar to OSSE1 except that it assimilated Sea 23 Surface Height (SSH) from two Wide-Swath Altimeter (2S) datasets instead of nadir altimeter data." – Why not show results from OSSE2? They would be valuable to put in context of the other results. Are the results very similar to those of one of the other OSSEs? Please point out which if that is the case.

P6 – 8 – "The SSH variance in the NR compares very favorably with real altimeter observations" – It would be helpful to see the ratio of the NR variance to the observed variance. Fig. 2 of Benkiran 2021 is very small, and there are areas where the NR has greater variability than DUACS and areas of less variability. The North Atlantic in the Irminger and Labrador Seas certainly is quite different. Rather than a qualitative statement of comparison, a plot of the ratio would be a quantitative comparison.

P6 – 11 – "SSH variance error" – This might be better worded as "variance of the SSH error" or "SSH error variance". I see the notation changing throughout the paper. It would be good to maintain consistency throughout.

Table 1 – The VAR* is the VarError divided by the variance of the NR if I understand correctly. However, the OSSE1 and OSSE3 values seem to be normalized differently. For example, 21.2/15.6=1.359, 14.1/10.1=1.396, 30.4/24.8 = 1.225, 21.3/17.0 = 1.253. Apparently I do not understand how these were computed. The text on page 6 should be more clear.

P6 – 38 – "compares the error variance of the different OSSEs for wavelengths smaller than 200 km" – This is a good way to evaluate the change in skill depending on feature sizes. My question is how these scales were extracted from the OSSE and nature runs. Was it by a spatial convolution filter, and if so what are the properties? I am wondering how sharp the filter cutoff may be and also how fields near land are filtered. Similarly, what filtering was used to separate time scales less than 20 days?

P7 – 21 – "Box D in Figure 5B" – There is not a box labeled as D on Figure 5b. There is a box labeled D on Figure 3. Boxes C and D in Fig 3 are located in the North Atlantic near the Iceland Basin and Irminger Sea. Figure 2 of Benkiran 2021 compares the nature run (left below) to Duacs (right below) showing much more variability in the nature run than in the observations in the areas of boxes C and D of Figure 3. If the model is over energetic, the impact of the satellite observations could be over estimated.

[Figure]

P 7 – 27 – "This is mainly due to the weak signal in these regions and the limited space/time sampling of the nadir altimeter constellation at these wavelengths." – I do not expect this is correct. Region A in Figure 3 is near the equator where spatial scales are typically quite large. The primary features are large eddies that are typically well resolved by 3 nadir altimeters.

P 7 – 30 – "The contribution of the swath altimeter data contributes to a clear reduction in the error spectra in all these regions for wavelengths larger than 50 km." – I do not reach that conclusion from Figure 10. Figure 10 area A indicates the 3N+2S deviating from the FR around 120 km scales. Area B is near 80 km, area C is near 50, and area D is near 70 km.

P 7 – 33 – "The reduction of the error at the different wavelengths (ERspec) is defined as the percentage of the error with respect to FR (OSSE0)." The comparison of error reduction from the FR to OSSE1 and OSSE3 is somewhat misleading. There certainly is reduction, though the FR error variance should be larger than the NR variance. The predominant processes affecting SSH variability are eddy instabilities. The processes that are deterministically forced by winds would be expected to be a much smaller fraction of the variance. If the eddy features in the FR are stochastically positioned relative to the features in the NR, the FR error variance can be up to twice the NR variance. This can mislead interpretation. For example, if an OSSE reduced error variance to half the error variance of the FR relative to the nature run, the OSSE could have no skill. The OSSE error variance could be equal to the NR variance.

It would be helpful to have maps of the FR and OSSE error variance divided by the NR error variance. That would more clearly demonstrate where the OSSE has skill and would still convey the same information on the errors in the OSSE relative to the FR. The plots of coherence in Figure 11 do help alleviate this problem to an extent. It should be addressed in with more clear plots.

Figure 11 plots C and D should have the horizontal axis range extended to show the point at which the black lines cross 0.5.

P 8 – 17 – "The calculation of this coherence was based on filtered SSH fields of scales greater than 500 km to avoid the impact of large-scale and high frequency signals on the results" – Does this imply scales greater than 500

km were removed before computing the time coherence?  On first reading I interpreted it the opposite manner.  More clear wording would help the reader.

P 8 – 29 – "This error is significantly reduced by assimilating the nadir altimeter data (black profile) compared 29 to the free model (FR, orange profile)" – Again, the variance of the FR – NR is misleading and will be larger than the variability of the NR.  It would be useful to plot the NR variance of temperature and salinity on these.  A similar comment pertains to Figure 14 of the velocity errors.  The NR variance should be plotted on Figure 14 as well.

Returning to an earlier comment, this is a point at which the details of the assimilation are very important.  It is not explained how the SSH translates to a subsurface temperature and salinity.  If the covariances are not accurate, this can be a source of the lack of impact of the swath data.  As was done with the SSH in Figures 10, 11, and 12, the spectra of the temperature and salinity are valuable to understand the scales at which skill is gained.  The general science community will be more interested in these variables than the SSH.

P 9 – 25 – "Surface current forecast errors should be equivalent to today's surface current analysis errors or alternatively will be improved (variance error reduction) by 30% at the surface and 50% for 300 m depths." – This is a confusing statement.  Why should the forecast errors be the equivalent to today's?  The results show the reductions.  We should expect reduction in forecast current errors.

Figure 15 – the Titles at the top of the panels seem to have an error.  Three of the panels are  titled "U Variance Error".

---

## Referee Comment (RC2)

Review of the paper "Contribution of a constellation of two Wide-Swath Altimetry Missions to Global Ocean Analysis and Forecasting" (os-2021-108)

General remarks:

The main purpose of this article is to assess the added value from assimilating altimeter data from two future wide-swath altimeters (WiSA) in a high-resolution global ocean forecasting system. The assessment is based on results from Observing System Simulation Experiments (OSSEs). The experimental results provide evidence that the wide-swath altimeters can substantially improve the analysis and forecast of mesoscale features.

The article is generally well written and easy to read. However, in view of the idealized nature of the experimental framework, the authors should be more nuanced in their discussion and conclusions regarding the expected impact of WiSA in operational data assimilation. A major simplification in the experiments is in the representation of WiSA observation errors, which is unrealistically simple compared to expected error sources in wide-swath altimetry, as modeled by the SWOT simulator. Only uncorrelated KaRIn noise is accounted for, with no justification as to why the other significant error sources (roll errors, phase errors…) are neglected. In particular, the roll and phase errors have a highly spatially correlated component, which needs to be adequately accounted for in the data assimilation system in order to be able to assimilate this high-resolution data-set effectively. The authors should at least recognize this important issue in the discussion, especially as this requires non-trivial developments to existing data assimilation systems.

Specific remarks:

1. Section 1, line 7. "*and has convinced more than thirty thousand expert services and users worldwide.*" Do the authors mean "*attracted*" instead of "*convinced*"?
2. Section 1, line 26. "*The main limitation of SWOT is, however, related to its long-time repeat period.*" What about the limitations of existing data assimilation systems to properly assimilate high-resolution SWOT observations?
3. Section 3.1, line 18. "*The second model is used to assimilate synthetic observations from the NR in a so-called Free Run (FR).*" A free run usually refers to a simulation that does not assimilate data, yet here we are told that it does assimilate data. If so, then which data are assimilated. Please clarify.
4. Section 3.2, p4, lines 33 until end of paragraph. "*The simulator models the most significant errors that are expected to affect the data… In this study, we only use the estimated WiSA KaRIn noise…*". Related to my main general remark above about the observation error specification, please provide more justification of this choice and discuss the implications.
5. Section 3.2. The noise level of WiSA is expected to be larger than that of SWOT (p3, last paragraph). Have the authors adjusted the SWOT simulator parameters to prescribe larger errors indicative of those of WiSA?
6. Section 3.4. "*OSSE2 (not presented here) is similar to OSSE1….*". If the results of OSSE2 are not presented then the authors can remove the reference to this experiment.

7. Section 4.1, line 11. "*The temporal evolution of the SSH variance error over the global ocean…*". It is unclear what is meant by "*variance error*". Is this the mean squared error (MSE); i.e., the global average of the squared differences between OSSE and nature run fields (with the mean removed)? A simple formula could help here. This is important as several diagnostics in the paper are based on this quantity. If it is the MSE then why present the squared errors instead of the root mean squared errors (RMSE) (or standard deviation), which is more common and easier to interpret since it has the same physical units as the field itself. It will affect the percentage error reductions reported in the paper; e.g., the reported reduction of 54% becomes 24% when considering the reduction of RMSE (or standard deviation).

8. Section 4.2, lines 37-38. "*…errors are characterized for specific time and space scales.*" Please give some detail on how the time and space scales have been separated. Presumably the authors are using a filter of some sort.

9. Section 4.2, p8, lines 17-18. "*…was based on filtered SSH fields…*". Please provide some detail on how the fields are filtered.

10. Section 5. "*Results confirm the high potential of such a configuration. Flying a constellation of two wide-swath altimeters will provide a major improvement…*". This is an idealized study so alternative wording should be used to be less definitive; e.g., "*Results suggest the high potential…*" and "*should provide a major improvement*". Proper assimilation of these observations will require effective data assimilation systems, beyond the current state of the art. More sophisticated treatment of observation errors (correlations and biases), improved background error covariances, and adequate treatment of model bias in data assimilation are important requirements in this respect. Uncertainty in the mean dynamic topography also remains a major issue for the assimilation of all forms of altimeter data (nadir as well as wide-swath).

11. Section 5, p9, line 25. "*Surface current forecast errors should be equivalent to today's surface current analysis errors…*". I don't understand this statement. Forecast errors with what lead time?

12. Many of the figure labels are difficult to read. Please use a larger font.

Minor corrections:

1. P1, line17. "point out" (?) instead of "recall"
2. P1, line 33. Remove "system".
3. P3, line 36. "What is the relationship between "a feature diameter" and "wavelength"?
4. P5, line 9. "*in situ*".
5. P5, line 14. "a free simulation".
6. P5, line 17. "model corrections".
7. P5, line17. "velocity field".
8. P5, line 26. "profile".
9. P6, line 6. "in Figure 3".
10. P6, line 25. "SST" (use previously defined acronym).
11. P6, line 31. "nadir" (not "Nadir" to be consistent with rest of article).
12. P6, line 33. "in the global".
13. P7, line 21. "Bonaduce et al. (2018)".
14. P8, line 6. "where" and "and" should not be in italics.
15. P8, line 6. "signals j where j refers to the experiment".

16. P8, line 38. "components." (missing period).
17. P15, Figure 4 caption. "black lines," and "altimeters".
18. P24, Figure 13 caption. "comparing temperature and salinity", not "zonal and meridional velocities".

---

## Author Comment (AC1)

Review of "Contribution of a constellation of two Wide-Swath Altimetry Missions to Global Ocean Analysis and Forecasting" by Mounir Benkiran, Pierre-Yves Le Traon, and Gérald Dibarboure

The manuscript is generally well written and understandable. The authors present results of several ocean prediction OSSEs to simulate the impact of two swath altimeter satellites in combination with three nadir altimeter satellites. The results are well presented. The results are significant and will be of interest to the science community for determining the importance of future investment. The quality of the work is high. The results are based on realistic ocean forecast systems being applied operationally. These results build on many prior studies on the impact of satellite and in situ observations. Most the comments I have are for improving the clarity of text and presentation. There are a few points that may require further consideration to resolve, though it is certainly possible I did not understand the intent of the text clearly, and in this case it may be relatively easy to resolve. The significant points are highlighted in yellow below.

7: "and has convinced more than thirty thousand expert services and users worldwide" – Do you mean that there are 30,000 users worldwide?

Yes more than 30,000 users (or intermediate users) worldwide

18: "so there is a strong diminishing return associated with classical altimetry" – we have seen a linear improvement in spatial correlation in properties associated with fronts. Surface divergence and frontogenesis forcing in particular increase linearly from 1 to 4 nadir altimeters. So the statement on diminishing returns depends on which properties are examined. For ssh, the statement is certainly correct. Clarification would be helpful.
Indeed, this applies to ssh. Sentence was modified as follows. "so there is a strong diminishing return for SSH mapping associated with classical altimetry. As a result…. Note that this statement depends on which properties are examined (see discussion in Jacobs et al., 2014) https://doi.org/10.1016/j.ocemod.2014.02.004

P3, 17 – "it is simply impossible to resolve the time scales of the small-scale features" That is a bit of a strong statement that could use some context. It applies to submesoscale instabilities in the open ocean. Arguably, tidal variability in coastal areas will be resolved in space, and though aliased in time SWOT observations should provide new information on small scale tides. Help the reader understand the particular ocean features to which the statement applies.

Was rephrased as follows: "It is simply impossible to resolve the time scales of the small mesoscale and submesocale turbulent motions"

P3, 20 – "can be observed with a mean revisit time of 5 days or less" – What is the source of the 5 day limit? Energetic areas such as the Gulf Stream or Kuroshio exhibit large changes even over 3 day time scales.

You are right but 5 days is already twice what we get along Jason/Sentinel 6 tracks. Observing in 2D 50 km features every 5 days would already be a major improvement compared to the present situation.

P3, 26 – "good enough for tidal aliasing despite being sun-synchronous" – That depends on application. It is not necessarily good enough for diurnal tides.

Of course a sun-synchronous orbit is not recommended for tidal studies. The point was just to state that reasonable sun synchroneous orbit choices can be made to reduce tidal aliasing problems.

P3, 35 – "spectral noise floor of 2 cm²/cycle/km" – The SWOT requirement should be stated clearly. Leaving the statement as this leaves the reader to assume this applies in the along track and cross track directions and that the noise is uncorrelated. The correlated error levels are very important. These have specific correlation shapes in the across track direction with spectral power that is not white noise in the along track direction.

This is right. The WISA error budget of the CNES 2020 reference follows the SWOT logic for the sake of consistency, with added nadir requirements for scales larger than 1000 km which SWOT does not have. In essence WISA combines the requirements of SWOT up to 1000 km and nadir altimeters for accuracy. It is right that correlated error can be quite complex and that random noise is not enough to give the full picture. This is captured in the Esteban Fernandez et al error document, and arguably not intuitive for many users. Discussing the details of the SWOT error budget is beyond the scope of the manuscript, but we tried to explain this logic in the revised version and to emphasize that random decorrelated noise is only *one* error component.

P4, 14 – "In our study, we use the same NEMO 1/12° resolution model but with different configurations and forcings" – It would be helpful for readers to have a few additional details on the model system at this point. What is the vertical resolution? This would help inform the reader as to the realism of the vertical structure of features in the nature run. What surface forcing is used at what temporal frequency? Is tidal forcing used in the system? What is the turbulence closure representation? Since the nature run is a free run, what is the realism of the circulation and energetics?

We have added a table in the text comparing the two models (NatRun vs FreeRun). In our study, no tidal forcing is used. See also Benkiran et al. (2021)

P4, 25 – "or Sentinel 6, which will use the same orbit" – verb tense. Sentinel-6A already is in the same orbit.

OK corrected

P4, 31 – "All SSH data were simulated from the NR using the Jet Propulsion Laboratory's (JPL) SWOT Simulator" – It was stated earlier that the WiSA design is simple and more cost efficient than SWOT though could resolve features only at larger scales. What is changed in the SWOT simulator to account for these differences? Are there any differences in correlated errors in the system? Will the predominant correlated errors still be baseline, roll, phase, and timing?

In this study we only took into account the non-correlated errors, i.e. the instrumental noise (which is larger than for SWOT). We took an observation error (figure 1 below) which depends on the resolution for a wave height of 2m. This error is larger than that of SWOT (Figure 7, Benkiran et al. 2021) for the same resolution (6 km x 6 km).

Generally speaking, other WiSA requirements are the same as for SWOT: the along-track PSD should be less than 10% of the SLA PSD up to 1000 km. Like for SWOT, correlated errors fall within these requirements: they are not specified above 1000 km except through the nadir requirements expressed in cm RMS. The SWOT error budget document (from Esteban Fernandez) describes in details the relative order of magnitude of most of the components: random noise dominates at scales below 50 km, then wet tropo error dominates near 70-200 km, then calibration residual progressively take over beyond this point. All these contributions are accounted for in the SWOT simulator, albeit with simplifications.

[Figure]

P4, 35 – "In this study, we only used the estimated WiSA KaRIn noise derived for a significant wave height (SWH) of 2 m" – How might correlated noise affect the results? These are significant in the roll, phase, timing, and baseline errors of SWOT. Correlated noise usually requires thinning or decimation of the data to the point where observation noise is uncorrelated, or the observation noise cross covariance matrix should account for this explicitly.

Difficult to quantify without dedicated studies and an end to end simulation. We have added the following paragraph in the paper conclusion to take into account this comment.

"Follow up studies should consider the full error spectrum taking into account, in particular, correlated long wavelength errors inherent to altimeter wide swath techniques (e.g. roll errors). This will require first to better specify these errors given the instrument and platform designs and to assess the impact of techniques that will be used to reduce them. As demonstrated by a series of studies carried out for the preparation of the SWOT mission (e.g. Dibarboure and Ubelmann, 2014), techniques such as swath / swath and swath / nadir cross over minimization will allow reducing a large part of these errors. We thus plan to carry out more advanced OSSEs that take into account the full error spectrum of wide swath altimeters, the reduction of these errors through cross-calibration techniques and the assimilation of corrected data and their residual (correlated) errors in advanced data assimilation schemes. "

Dibarboure, G., & Ubelmann, C. (2014). Investigating the Performance of Four Empirical Cross-Calibration Methods for the Proposed SWOT Mission. Remote Sensing, 6(6), 4831-4869.

P5 – 17 – "All these updates and their impacts on the system performance are described in Benkiran et al. (2021)." – Details would help the reader here. There is some contradiction with the information on lines 3-4 of page 5 stating a 7 day assimilation cycle versus the statement of 1 day. Though details are in Benkiran, it would be helpful to have some basic information here. What is the frequency at which the analysis is performance (daily or 7 days)? What is the data interval for each analysis (the prior 24 hours, prior 7 days, a 7 day period centered on the analysis time, …)? Is the data thinned or decimated in the processing for the analysis? For a 1/12 degree model that represents features on the order of 45 – 90 km wavelength, the 6 km sampling of the WiSA data produces greatly redundant information. How is this handled? If the assimilation system uses a covariance produced from model free run, and the observations are from a model free run, then the simulated data will be very consistent with the assimilation system assumptions within the covariances. How will this affect the results in the real world if the assumed covariances are not accurate? Will a WiSA still produces the same level of impact concluded in the paper?

Yes, all the updates are described in the arctile (Benkiran et al., 2021). I give you some information, the system uses a 7-day analysis (7 days of data), a mid-cycle analysis and generates daily increments (corrections). As detailed in Benkiran et al, this analysis uses daily forecast error covariance. Yes, we have 6km resolution for the data (Wisa, Nadirs) and we have implemented an algorithm in the system that removes the redundant data in the analysis bubble. This allowed us to remove redundant data and reduce the number of observations in the analysis bubble. The free system from which the data is simulated is not the same as the one from which we calculated our covariances. We calibrated our OSSEs with an OSE (real data), example in the paper Benkiran et al. ,2021. This calibration brought us closer to the real world where the covariances used are not exact. Yes WiSA will give the same level of impact because the Nadirs data will use the same covariances and the same system configuration.

P5 – 23 – "OSSE2 (not presented here) is similar to OSSE1 except that it assimilated Sea Surface Height (SSH) from two WiSA (Wide-Swath) Altimeter (2S) datasets instead of nadir altimeter data." – Why not show results from OSSE2? They would be valuable to put in context of the other results. Are the results very similar to those of one of the other OSSEs? Please point out which if that is the case.

Yes, as the main objective is to assimilate both Nadirs and swath data, we have shown that this combination. Ok, we will remove this from the text. OSSE2 results are very close to OSSE3 results. For information, we present in the following figure the evolution of the error variance of these three experiments with OSSE2 in blue.

[Figure]

P6 – 8 – "The SSH variance in the NR compares very favorably with real altimeter observations" – It would be helpful to see the ratio of the NR variance to the observed variance. Fig. 2 of Benkiran 2021 is very small, and there are areas where the NR has greater variability than DUACS and areas of less variability. The North Atlantic in the Irminger and Labrador Seas certainly is quite different. Rather than a qualitative statement of comparison, a plot of the ratio would be a quantitative comparison.

As requested, we plotted the ratio between the variance of the NR and the variance of the observations (Duacs) in the following figure. We note that this ratio remains close to 1 except in some regions and in particular in the North Atlantic where the NR has too high a variance compared to the data. Not the Duacs data (produced by objective analysis) underestimates the variance, in particular, in high latitude regions and filter high frequency signals.

[Figure]

*Ratio of the NR variance to the observed variance (2015)*

P6 – 11 – "SSH variance error" – This might be better worded as "variance of the SSH error" or "SSH error variance". I see the notation changing throughout the paper. It would be good to maintain consistency throughout.

We have corrected by " SSH error variance" in the text

Table 1 – The VAR* is the VarError divided by the variance of the NR if I understand correctly. However, the OSSE1 and OSSE3 values seem to be normalized differently. For example, 21.2/15.6=1.359, 14.1/10.1=1.396, 30.4/24.8 = 1.225, 21.3/17.0 = 1.253. Apparently I do not understand how these were computed. The text on page 6 should be more clear.

There is actually a confusion in this table and the legend (Table 2: SSH ocean analysis and forecast error statistics during the year 2015. Columns 1 and 2 represent the analysis and forecast variance of error computed from the difference between the OSSE and the NR (VarError, cm$^2$). Columns 3 and 4 show the OSSEs error variance relative the FR error variance (Var*, %)).

 The values were corrected and the calculation is detailed:
      VarError(FR) : SSH variance error of the free simulation (FR) is about 73.4cm$^2$ (map average: figure 5A).

      Var* (%) = 100.*VarError(OSSEs) / VarError(FR), which gives the relative gain with respect to the FR. We have added more detail in the article with an expression of this normalized error.

P6 – 38 – "compares the error variance of the different OSSEs for wavelengths smaller than 200 km" – This is a good way to evaluate the change in skill depending on feature sizes. My question is how these scales were extracted from the OSSE and nature runs. Was it by a spatial convolution filter, and if so what are the properties? I am wondering how sharp the filter cutoff may be and also how fields near land are filtered. Similarly, what filtering was used to separate time scales less than 20 days?

For filtering (temporal and spatial), we used python functions (scipy.signal.bessel). Cut-off parameters were defined according to the spectral slope . A spectral study was carried out to validate this cut-off. We use an extrapolation of the field strength near the coast taking into account the distance to the coast.

P7 – 21 – "Box D in Figure 5B" – There is not a box labeled as D on Figure 5b. There is a box labeled D on Figure 3. Boxes C and D in Fig 3 are located in the North Atlantic near the Iceland Basin and Irminger Sea. Figure 2 of Benkiran 2021 compares the nature run (left below) to Duacs (right below) showing much more variability in the nature run than in the observations in the areas of boxes C and D of Figure 3. If the model is over energetic, the impact of the satellite observations could be over estimated.

The figure number is corrected in the text (figure 3).
Yes, as described before, the NR has a higher variability than in the observations (Duacs) in the North Atlantic. As shown in the figure below (Var(NR)/Var(FR)), the NR is close to the FR in this region. In the paper by benkiran et al, 2021, (Supplementary Figure 1 : Spatial distribution of Standard deviation of Sea Level Anomaly (SLA) error: (A) G12, (B) OSSE system, and (C) Scatter diagram between G12 and OSSE Standard deviation of SLA Error.), we compared the error of our OSSE with that of the model that assimilates the real data (G12: CMEMS GLORYS12 reanalysis). We showed that these errors are of the same order. Note also that the observations (Duacs maps) certainly underestimate variability here.

P 7 – 27 – "This is mainly due to the weak signal in these regions and the limited space/time sampling of the nadir altimeter constellation at these wavelengths." – I do not expect this is correct. Region A in Figure 3 is near the equator where spatial scales are typically quite large. The primary features are large eddies that are typically well resolved by 3 nadir altimeters.

Yes, you are right, on this box (near the equator) we have large structures that can be well resolved by the Nadirs data, but in our case, the (2D) wide-swath data assimilation also contains the large scale better than the Nadirs altimeters. The spatio-temporal coverages of the swath data are more homogeneous than those of the Nadirs during our assimilation cycle (7 days). In this region, fast phenomena dominate, and the assimilation of Nadirs altimeter observations has difficulties to properly describe the fast mesoscale.

P 7 – 30 – "The contribution of the swath altimeter data contributes to a clear reduction in the error spectra in all these regions for wavelengths larger than 50 km." – I do not reach that conclusion from Figure 10. Figure 10 area A indicates the 3N+2S deviating from the FR around 120 km scales. Area B is near 80 km, area C is near 50, and area D is near 70 km.

The contribution of the swath altimeter data contributes to a clear reduction in the wavelength error spectra between 50 and 100 km depending on the latitude.

P 7 – 33 – "The reduction of the error at the different wavelengths (ERspec) is defined as the percentage of the error with respect to FR (OSSE0)." The comparison of error reduction from the FR to OSSE1 and OSSE3 is somewhat misleading. There certainly is reduction, though the FR error variance should be larger than the NR variance. The predominant processes affecting SSH variability are eddy instabilities. The processes that are deterministically forced by winds would be expected to be a much smaller fraction of the variance. If the eddy features in the FR are stochastically positioned relative to the features in the NR, the FR error variance can be up to twice the NR variance. This can mislead interpretation. For example, if an OSSE reduced error variance to half the error variance of the FR relative to the nature run, the OSSE could have no skill. The OSSE error variance could be equal to the NR variance.

It would be helpful to have maps of the FR and OSSE error variance divided by the NR error variance. That would more clearly demonstrate where the OSSE has skill and would still convey the same information on the errors in the OSSE relative to the FR. The plots of coherence in Figure 11 do help alleviate this problem to an extent. It should be addressed in with more clear plots.

Figure 11 plots C and D should have the horizontal axis range extended to show the point at which the black lines cross 0.5.

In the following figure, we present (as requested) the ratio between the FR error variance and the NR variance, we have values around 1 except in regions of high variability (GS, ACC...).

In the following figure, we have the ratio of the SSH Variance of the FR to the NR. We can see that the two simulations have quite different variabilities, with quite different structures, positions of the currents in the two simulations.

We redid the plots of the temporal spectrum with a temporal extension. It is clearer now.

[Figure]

[Figure]

[Figure]

P 8 – 17 – "The calculation of this coherence was based on filtered SSH fields of scales greater than 500 km to avoid the impact of large-scale and high frequency signals on the results" – Does this imply scales greater than 500 Km were removed before computing the time coherence? On first reading I interpreted it the opposite manner. More clear wording would help the reader

Yes, scales greater than 500 km were removed. Sentence was rewritten as follows "To remove the impact of large-scale and high frequency signals, a high pass filter to remove scales larger than 500 km was applied to the SSH fields before computing the time coherence".

P 8 – 29 – "This error is significantly reduced by assimilating the nadir altimeter data (black profile) compared 29 to the free model (FR, orange profile)" – Again, the variance of the FR – NR is misleading and will be larger than the variability of the NR. It would be useful to plot the NR variance of temperature and salinity on these. A similar comment pertains to Figure 14 of the velocity errors. The NR variance should be plotted on Figure 14 as well. Returning to an earlier comment, this is a point at which the details of the assimilation are very important. It is not explained how the SSH translates to a subsurface temperature and salinity. If the covariances are not accurate, this can be a source of the lack of impact of the swath data. As was done with the SSH in Figures 10, 11, and 12, the spectra of the temperature and salinity are valuable to understand the scales at which skill is gained. The general science community will be more interested in these variables than the SSH.

As shown in the paper, the impact of assimilating swath altimetry has a positive impact on temperature and salinity. The following figures show the spectral coherences for the same boxes presented in the paper for SHH but here for SST and SSS. There is indeed a better resolution with the assimilation of swath data, in particular, for SST.

[Figure]

*Sea Surface Temperature Wavenumber spectral coherence with respect to the NR, (A) low-latitude region (red box in Figure 3), (B) Agulhas current (orange box in Figure 3), (C) North Atlantic (high-latitude, green box in Figure 3) and (D) North Atlantic Drift current (black box in Figure 3).*

[Figure]

*Sea Surface Salinity  Wavenumber spectral coherence with respect to the NR, (A) low-latitude region (red box in Figure 3), (B) Agulhas current (orange box in Figure 3), (C) North Atlantic (high-latitude, green box in Figure 3) and (D) North Atlantic Drift current (black box in Figure 3).*

P 9 – 25 – "Surface current forecast errors should be equivalent to today's surface current analysis errors or alternatively will be improved (variance error reduction) by 30% at the surface and 50% for 300 m depths." – This is a confusing statement. Why should the forecast errors be the equivalent to today's? The results show the reductions. We should expect reduction in forecast current errors.

The point we made is that future 7-day forecasts (that could benefit from the assimilation of two swath altimeters) will be as accurate as our today analyses (derived from the assimilation of nadir altimeters). The phrase was reformulated.

"Surface current forecast errors will be improved (variance error reduction) by 30% at the surface and 50% for 300 m depths.  These forecast errors should be equivalent to today's analysis errors derived from nadir altimeters. "

Figure 15 – the Titles at the top of the panels seem to have an error. Three of the panels are titled "U Variance Error". Done

---

## Author Comment (AC2)

Review of the paper "Contribution of a constellation of two Wide-Swath Altimetry Missions to Global Ocean Analysis and Forecasting" (os-2021-108)

General remarks

The article is generally well written and easy to read. However, in view of the idealized nature of the experimental framework, the authors should be more nuanced in their discussion and conclusions regarding the expected impact of WiSA in operational data assimilation. A major simplification in the experiments is in the representation of WiSA observation errors, which is unrealistically simple compared to expected error sources in wide-swath altimetry, as modeled by the SWOT simulator. Only uncorrelated KaRIn noise is accounted for, with no justification as to why the other significant error sources (roll errors, phase errors...) are neglected. In particular, the roll and phase errors have a highly spatially correlated component, which needs to be adequately accounted for in the data assimilation system in order to be able to assimilate this high-resolution data-set effectively. The authors should at least recognize this important issue in the discussion, especially as this requires non-trivial developments to existing data assimilation systems.

Specific remarks:

1. Section1,line7."*and has convinced more than thirty thousand expert services and users worldwide.*" Do the authors mean "*attracted*" instead of "*convinced*"?

   Rephrased as "and is now used by more than thirty thousand expert services and users worldwide"

2. Section 1, line 26. "*The main limitation of SWOT is, however, related to its long-time repeat period.*" What about the limitations of existing data assimilation systems to properly assimilate high-resolution SWO observations?

   There are indeed limitations in the data assimilation systems but the main limitation is related to the time sampling. Note that in our SWOT OSSE studies, we made several improvements in the data assimilation system (see Benkiran et al, 2021). Other improvements could be done. In particular, we kept a time window (analysis cycle length) of 7 days to stay as close as possible to our operational system (that assimilates many types of data) while a 10-day cycle (the SWOT sub-cycle) would have been preferrable.

3. Section 3.1, line 18. "*The second model is used to assimilate synthetic observations from the NR in a so-called Free Run (FR).*" A free run usually refers to a simulation that does not assimilate data, yet here we are told that it does assimilate data. If so, then which data are assimilated. Please clarify.

   The second model called Free-Run (Control Run) is a model without data assimilation. This experiment is called OSSE0 in the following. However, it is with this system that we will do our experiments, we will assimilate the pseudo-data and the experiments will be named OSSE1 and OSSE3 in the following.

4. Section 3.2, p4, lines 33 until end of paragraph. "*The simulator models the most significant errors that are expected to affect the data... In this study, we only use the estimated WiSA KaRIn noise...*". Related to my main general

remark above about the observation error specification, please provide more justification of this choice and discuss the implications.

The study was indeed a first step. We have added this paragraph in the conclusion that answers the reviewer's concern:

Follow up studies should consider the full error spectrum taking into account, in particular, correlated long wavelength errors inherent to altimeter wide swath techniques (e.g. roll errors). This will require first to better specify these errors given the instrument and platform designs and to assess the impact of techniques that will be used to reduce them. As demonstrated by a series of studies carried out for the preparation of the SWOT mission (Dibarboure et al., 2014), techniques such as swath / swath and swath / nadir cross over minimization will allow reducing a large part of these errors. We thus plan to carry out more advanced OSSEs that take into account the full error spectrum of wide swath altimeters, the reduction of these errors through cross calibration techniques and the assimilation of corrected data and their residual (correlated) errors in advanced data assimilation schemes.

5. Section 3.2. The noise level of WiSA is expected to be larger than that of SWOT (p3, last paragraph). Have the authors adjusted the SWOT simulator parameters to prescribe larger errors indicative of those of WiSA?

Yes the error level of Wisa is higher than SWOT. The simulator includes an error file that depends on the type of data we want to simulate. Here, for the Wisa study, we used specific errors from which the simulation calculates the Karin noise and the errors of the data with the requested data resolution.

6. Section3.4.“OSSE2(not presented here )is similar to OSSE1....”.If the results of OSSE2 are not presented then the authors can remove the reference to this experiment.

Yes, as the main objective is to assimilate both Nadirs and swath data, we have shown that this combination. Ok, we will remove this from the text. For information, we present in the following figure the evolution of the error variance of these three experiments with OSSE2 in blue.

[Figure]

7. Section4.1,line11.“*The temporal evolution of the SSH variance error over the global ocean*...”. It is unclear what is meant by “*variance error*”. Is this the mean squared error (MSE); i.e., the global average of the squared differences between OSSE and nature run fields (with the mean removed)? A simple formula could help here. This is important as several diagnostics in the paper are based on this quantity. If it is the MSE then why present the squared errors instead of the root mean squared errors (RMSE) (or standard deviation), which is more common and easier to interpret since it has the same physical units as the field itself. It will affect the percentage error reductions reported in the paper;

e.g., the reported reduction of 54% becomes 24% when considering the reduction of RMSE (or standard deviation).

Here we use the error variance (difference between NatRun and OSSEs) as follows:

$$\text{VAR Error}(\text{OSSE}_j) = \frac{1}{T}\sum_{t=0}^{T}\left(\Delta SSH(t) - \overline{\Delta SSH(t)}\right)^2$$

$$\Delta SSH = \text{SSH}(OSSE_j) - \text{SSH (NR)}$$

is the temporal variance of SSH error obtained by comparing the NR (NatRun) with a given OSSE at a given location x and y over a period of t 363 days with j 1; 2 referring to the j-th OSSE and nt referring to the maximum time.

8. Section4.2,lines 37-38."...*errors are characterized for specific time and space scales.*" Please give some detail on how the time and space scales have been separated. Presumably the authors are using a filter of some sort.

for filtering (temporal and spatial) we used python functions (scipy.signal.bessel), we set the cut-off parameters based on spectral slopes.

9. Section 4.2, p8, lines 17-18. "...*was based on filtered SSH fields...*". Please provide some detail on how the fields are filtered.

In a first step we calculated the temporal coherences with the total SSH signal, we obtained (following figure on the right) noisy coherences at small temporal scales (around 10 days). In a second step we did the same operation with the different spatial scale games and we found that this noise comes from the large scale high frequency structures. as it is not our objective here, we decided to filter this large scale high frequency spatial scale to get the following figure on the left.

[Figure]

[Figure]

10. Section 5. "*Results confirm the high potential of such a configuration. Flying a constellation of two wide-swath altimeters will provide a major improvement...*". This is an idealized study so alternative wording should be used to be less definitive; e.g., "*Results suggest the high potential...*" and "*should provide a major improvement*". Proper assimilation of these observations will require

effective data assimilation systems, beyond the current state of the art. More sophisticated treatment of observation errors (correlations and biases), improved background error covariances, and adequate treatment of model bias in data assimilation are important requirements in this respect. Uncertainty in the mean dynamic topography also remains a major issue for the assimilation of all forms of altimeter data (nadir as well as wide-swath).

We agree and modified the text as proposed (suggest the high potential and should provide). See also answer to comment 4

11. Section 5, p9, line 25. "*Surface current forecast errors should be equivalent to today's surface current analysis errors*...". I don't understand this statement. Forecast errors with what lead time?

7-day .  This is precised now.  Sentence was also rewritten

12. Many of the figure labels are difficult to read. Please use a larger font.

All figures were improved with a larger (clearer) label

Minor corrections:

1. P1,line17."point out"(?)insteadof"recall" done
2. P1,line33.Remove"system". Done (P2, line 33)
3. P3, line 36. "What is the relationship between "a feature diameter" and "wavelength" the wavelength is twice the diameter
4. P5,line9."*insitu*". done
5. P5,line14."afreesimulation". done
6. P5,line17."modelcorrections". done
7. P5,line17."velocityfield". done
8. P5,line26."profile". done
9. P6,line6."inFigure3". done
10. P6, line 25. "SST" (use previously defined acronym). done
11. P6, line 31. "nadir" (not "Nadir" to be consistent with rest of article). 12.P6, line 33. "in the global". done
13. P7, line 21. "Bonaduce et al. (2018)". done
14. P8, line 6. "where" and "and" should not be in italics. done
15. P8, line 6. "signals j where j refers to the experiment". done

1.

---

## Referee Report (RR1)

Review of "Contribution of a constellation of two Wide-Swath Altimetry Missions to Global Ocean Analysis and Forecasting" by Mounir Benkiran, Pierre-Yves Le Traon, and Gérald Dibarboure

The manuscript is generally well written and understandable. The authors present results of several ocean prediction OSSEs to simulate the impact of two swath altimeter satellites in combination with three nadir altimeter satellites. The results are well presented. The results are significant and will be of interest to the science community for determining the importance of future investment. The quality of the work is high. The results are based on realistic ocean forecast systems being applied operationally. These results build on many prior studies on the impact of satellite and in situ observations.

The authors' response to comments is very good, and I appreciated understanding the explanations provided. The explanations helped me understand the details more clearly and put the results into context, and the changes to the text helped me avoid accidental misinterpretation. I have only a couple minor comments on the revised manuscript:

Page 5: "OSSE0 is the Free Run (FR) of the ocean model used to assess the performance of the other experiments. OSSE1 corresponds to nadir (3N) altimetry data assimilation. Finally, OSSE3 (3N+2S) assimilated all observation types (combining two swaths and three nadirs). OSSE1, 2 and 3 also assimilate Sea Surface Temperature (SST), Ice Concentration (IC), and Temperature and Salinity (T/S) profile data." – Check the number of the OSSEs throughout. In these two sentences the numbering is different.

Page 8: This should be equation 2, and there should be a square the numerator.

---

## Referee Report (RR2)

Review of the revised paper "Contribution of a constellation of two Wide-Swath Altimetry Missions to Global Ocean Analysis and Forecasting" (os-2021-108)

General remarks:

Apart from the specific remarks below, the authors have done an adequate job of revising the paper in response to my remarks. In responding to the reviewers, the authors should have highlighted modified passages with a different color and indicated explicitly in the response to each remark whether they modified the text, giving page and line numbers. This is standard practice for helping the task of the reviewer.

Specific remarks:

1. Abstract. "*Sea Surface Height (SSH) analysis and 7-day forecast error will be globally reduced by about 50%.*" This is speculation. All you can do here is report what you've seen in your OSSEs, which are idealized experiments that are likely giving overly optimistic results. Replace "*will be*" by "*are*" and then for clarity add "*in the OSSEs.*" at the end of the sentence. This remark also applies to the first paragraph in the Summary and Conclusions.
2. Response to Remark 3: Section 3.1. The sentence in the paper is unaltered and the explanation still leaves me confused. It may be the choice of wording that's confusing me. A "Run" usually refers to a simulation, not the model itself. Here, "Free Run" seems to refer to a particular model configuration that's used for simulations with and without data assimilation. But later on (first paragraph Section 3.4), we're told that OSSE0 (no assimilation) is the Free Run. Why not just say: "*The second model is used to assimilate synthetic observations from the NR. This model uses…*". And then explain the Free Run (no data assimilation) when you describe the OSSEs.
3. Response to Remark 7. This is clear now. These equations and explanation should be included in the paper, after equation (1) so it's completely clear what you're presenting.

---

## Author Response (AR2)

Review 1 :

Review of "Contribution of a constellation of two Wide-Swath Altimetry Missions to Global Ocean Analysis and Forecasting" by Mounir Benkiran, Pierre-Yves Le Traon, and Gérald Dibarboure The manuscript is generally well written and understandable. The authors present results of several ocean prediction OSSEs to simulate the impact of two swath altimeter satellites in combination with three nadir altimeter satellites. The results are well presented. The results are significant and will be of interest to the science community for determining the importance of future investment. The quality of the work is high. The results are based on realistic ocean forecast systems being applied operationally.

These results build on many prior studies on the impact of satellite and in situ observations. The authors' response to comments is very good, and I appreciated understanding the explanations provided. The explanations helped me understand the details more clearly and put the results into context, and the changes to the text helped me avoid accidental misinterpretation. I have only a couple
minor comments on the revised manuscript:

Page 5: "OSSE0 is the Free Run (FR) of the ocean model used to assess the performance of the other experiments. OSSE1 corresponds to nadir (3N) altimetry data assimilation. Finally, OSSE3 (3N+2S) assimilated all observation types (combining two swaths and three nadirs). OSSE1, 2 and 3 also assimilate Sea Surface Temperature (SST), Ice Concentration (IC), and Temperature and Salinity (T/S) profile data." – Check the number of the OSSEs throughout. In these two sentences the numbering is different. done

Page 8: This should be equation 2, and there should be a square the numerator. done

Review 2:

Review of the revised paper "Contribution of a constellation of two Wide-Swath Altimetry Missions to Global Ocean Analysis and Forecasting" (os-2021-108)
General remarks:
Apart from the specific remarks below, the authors have done an adequate job of revising the paper in response to my remarks. In responding to the reviewers, the authors should have highlighted modified passages with a different color and indicated explicitly in the response to each remark whether they modified the text, giving page and line numbers. This is standard practice for helping the task of the reviewer.

Specific remarks:
1. Abstract. "Sea Surface Height (SSH) analysis and 7-day forecast error will be globally reduced by about 50%." This is speculation. All you can do here is report what you've seen in your OSSEs, which are idealized experiments that are likely giving overly optimistic results. Replace "will be" by "are" and then for clarity add "in the OSSEs." at the end of the sentence. This remark also applies to the first paragraph in the Summary and Conclusions.**Done**

2. Response to Remark 3: Section 3.1. The sentence in the paper is unaltered and the explanation still leaves me confused. It may be the choice of wording That's confusing me. A "Run" usually refers to a simulation, not the model itself. Here, "Free Run" seems to refer to a particular model configuration that's used for simulations with and without data assimilation. But later on (first paragraph Section 3.4), we're told that OSSE0 (no assimilation) is the Free Run. Why not just say: "The second model is used to assimilate synthetic observations from the NR. This model uses…". And then explain the Free Run (no data assimilation) when you describe the OSSEs. **Done**

3. Response to Remark 7. This is clear now. These equations and explanation should be included in the paper, after equation (1) so it's completely clear what you're presenting. **Done**